# On the Pleiotropic Actions of Glucagon-like Peptide-1 in Its Regulation of Homeostatic and Hedonic Feeding

**DOI:** 10.3390/ijms26083897

**Published:** 2025-04-20

**Authors:** Sarah Sayers, Ed Wagner

**Affiliations:** College of Osteopathic Medicine of the Pacific, Western University of Health Sciences, Pomona, CA 91766, USA; sarah.sayers@westernu.edu

**Keywords:** GLP1, dopamine, NPY, PACAP, TRPC5 channels, appetitive behavior, ventromedial nucleus

## Abstract

We examined the neuroanatomical substrates and signaling mechanisms underlying the suppressive effect of GLP1 on homeostatic and hedonic feeding. Electrophysiological and behavioral studies were conducted in agouti-related peptide (AgRP)-cre and tyrosine hydroxylase (TH)-cre mice, and AgRP-cre/pituitary adenylyl cyclase-activating polypeptide (PACAP) type I receptor (PAC1R)^fl/fl^ animals. GLP1 (30 pmol) delivered directly into the arcuate nucleus (ARC) decreased homeostatic feeding and diminished the rate of consumption. This anorexigenic effect was associated with an inhibitory outward current in orexigenic neuropeptide Y (NPY)/AgRP neurons. GLP1 injected into the ventral tegmental area reduced binge feeding, coupled with decrements in the rate of consumption and the percent daily caloric consumption during the binge interval. These reductions were associated with a GLP1-induced outward current in mesolimbic (A_10_) dopamine neurons. GLP1 administered into the ventromedial nucleus (VMN) reduced homeostatic feeding that again was associated with a diminished rate of consumption and abrogated by the GLP1 receptor antagonist exendin 9–39 and in AgRP-cre/PAC1R^fl/fl^ mice. This suppressive effect was linked with a GLP-induced inward current in VMN PACAP neurons, and further supported by the fact that GLP1 neurons in the nucleus tractus solitarius project to the VMN. Conversely, intra-VMN GLP1 had modest effects on binge feeding behavior. Finally, apoptotic ablation of VMN PACAP neurons obliterated the anorexigenic effect of intra-VMN GLP1 on homeostatic feeding in PACAP-cre mice but not their wildtype counterparts. Collectively, these data demonstrate that GLP1 acts within the homeostatic and hedonic circuits to curb appetitive behavior by exciting PACAP neurons, and inhibiting NPY/AgRP and A_10_ dopamine neurons.

## 1. Introduction

Glucagon-like peptide (GLP)1 is an incretin that is secreted postprandially from intestinal L cells in response to neural stimulation from the vagus and nutrient signals in the lumen, which results in enhanced glucose-dependent insulin secretion [1]. In doing so, it helps maintain glucose homeostasis and GLP1 analogues like liraglutide lessen the clinical impact of obesity and associated comorbidities [2]. GLP1 and its analogues decrease both homeostatic (i.e., ad libitum) and hedonic (i.e., palatability-based) feeding [3,4,5]. GLP1 receptors are located in hypothalamic areas like the arcuate nucleus (ARC), paraventricular nucleus (PVN) and ventromedial nucleus (VMN) [6], which regulate homeostatic feeding [7], as well as in areas controlling hedonic feeding (i.e., the ventral tegmental area (VTA) and nucleus accumbens [8]. Additionally, GLP1 neurons in the nucleus tractus solitarius (NTS) project to regions within both homeostatic and hedonic energy balance circuits [5,8].

Within the ARC, GLP1 receptors are expressed in proopiomelanocortin (POMC) neurons in a distinct yet overlapping pattern compared to leptin receptors [9]. These GLP1 receptor-expressing POMC neurons have a differing array of intrinsic conductances and are more inherently excitable [9]. In addition, a greater percentage of them respond to GLP1 with depolarization and increased firing than leptin receptor-expressing POMC neurons do in response to leptin [9]. In neuropeptide Y (NPY)/agouti-related peptide (AgRP) neurons, GLP1 is reported to inhibit these cells via a combination of presynaptic facilitation of γ-aminobutyric acid (GABA) release and a direct hyperpolarization via activation of ATP-gated K^+^ (K_ATP_) channels [10]. Emerging evidence also suggests that GLP1 activates transient receptor potential (TRP)C5 channels. For example, the GLP1-induced increase in excitability of POMC neurons is mimicked by its liraglutide analogue [10], and selective knockdown of TRPC5 channels in POMC neurons negates the depolarizing effect of liraglutide as well as the anorexigenic effect elicited by systemic injection [10]. On the other hand, knockdown of the GLP1 receptor in POMC or Sim1 neurons reportedly does not alter energy intake or expenditure, or the anorexigenic effect of GLP1 analogs [11].

While the actions of GLP1 within the ARC have been characterized at least partially, much less is known about its cellular effects in the VMN and VTA. A major anorexigenic population of pituitary adenylyl cyclase-activating polypeptide (PACAP) neurons is found in the VMN, and these cells also express steroidogenic factor-1 and glutamate [12,13,14,15]. VMN PACAP neurons inhibit both homeostatic and hedonic feeding [15,16], and they project to both the ARC and the VTA [16,17]. Given the critical role of VMN PACAP neurons in regulating appetitive behavior, there exists an unmet need to determine whether VMN PACAP neurons are the direct target of GLP1 action, and whether the GLP-1-induced excitation of these cells in turn inhibits ARC NPY/AgRP and VTA mesolimbic A_10_ dopamine neurons to decrease homeostatic and hedonic feeding, respectively. Thus, we tested the hypothesis that GLP1 excites VMN PACAP neurons via activation of TRPC5 channels that leads to downstream inhibition of ARC NPY/AgRP and VTA A_10_ dopamine neurons to account for its anorexigenic effect. To this end, we investigated site-specific electrophysiological and behavioral actions of GLP1 in the ARC, VMN and VTA, and addressed the question of whether VMN PACAP neurons and PACAP-selective PAC1 receptors expressed in NPY/AgRP neurons play a role in the effects of GLP1 on homeostatic and hedonic feeding.

## 2. Results

### 2.1. GLP1 Attenuates Feeding in the ARC, Likely Due to GLP1-Induced Inhibition of ARC NPY/AgRP Neurons

As previously mentioned, GLP1 receptors are located on ARC POMC neurons [9]. Seeing as the ARC mediates homeostatic feeding [7], and anorexigenic POMC neurons contain GLP1 receptors [9], we first tested the hypothesis that intra-ARC administration of GLP1 would reduce homeostatic appetitive behavior. The data show intra-ARC injection of GLP1 (30 pmol, 0.2 μL) significantly reduced energy intake at four and eight after administration (Figure 1A: repeated measures multi-factorial ANOVA/LSD: *F*_GLP1_: 36.10, DF: 1, *p*  <  0.0001; *F*_time_: 150.24, DF: 3, *p* < 0.0001; *F*_interaction_: 6.55, DF: 3, *p* < 0.0004; one-way ANOVA/LSD, *F*: 74.84, DF: 7, *p* < 0.0001, *n* = 6), without a change in bout duration (Figure 1B: repeated measures multi-factorial ANOVA/LSD: *F*_GLP1_: 0.01, DF: 1, *p <* 0.91; *F*_time:_ 0.38, DF: 3, *p* < 0.77; *F*_interaction_: 0.42, DF: 3, *p* < 0.74, *n* = 6), and a significant decrease in rate of consumption at all time points analyzed (Figure 1C: repeated measures multi-factorial ANOVA/LSD: *F*_GLP1_: 10.11, DF: 1, *p* < 0.002; *F*_time_: 1.77, DF: 3, *p* < 0.16; *F*_interaction_: 0.34, DF: 3, *p* < 0.80, *n* = 6). Although GLP1 treatment elicited a 0.41 g drop in body weight relative to saline-treated controls, this did not reach statistical significance (student’s *t*-test: t = 2.683, *p* > 0.05, *n* = 16). In order to further understand the mechanism behind these behavioral changes, we analyzed the effect of GLP1 on ARC neurons. We used visualized whole-cell patch clamp recordings of orexigenic ARC NPY/AgRP neurons from AgRP-cre mice visualized with eYFP (pAAV-Ef1a-DIO EYFP; Figure 2A–C). GLP1 (100 nM) promoted a robust and reversible outward current (Figure 2D). This outward current was completely reversed when GLP1 was administered in the presence of PI3K-Kinase inhibitor PI828 (10 μM) (Figure 2E), diminished in the presence of PLC inhibitor U73122 (20 μM) (Figure 2F), and abrogated when co-administered with K_ATP_ channel blocker tolbutamide (100 μM; Figure 2G). Composite data depicts PI828, U73122, and tolbutamide all significantly hindered the GLP1 induced positive change in current (Figure 2H: one-way ANOVA/LSD, *F*: 25.89, DF: 36, *p <* 0.0001: n = 7–13 cells, 7–9 ARC slices, 3–4 animals). The I/V plot shown in Figure 2I reveals that the GLP1 response reversed polarity at −90 mV, corresponding with the Nernst equilibrium potential for potassium conductance. These data depict that GLP1 promotes decreased appetitive behavior in part by inhibition of ARC NPY/AgRP neurons, likely through activation of PI3K–PLC–PKC signal transduction and K_ATP_ channels.

### 2.2. GLP1 Inhibits Hedonic Feeding via Inhibition of VTA A_10_ Dopamine Neurons

In order to further explore the effects of GLP1 on hedonic feeding behavior, we performed intra-VTA injections of GLP1 or its saline vehicle prior to subjecting animals to a one-hour exposure to HFD. GLP1 significantly reduced binge feeding with no significant difference in energy intake during the 23-h post-binge monitoring period (Figure 3A: Binge hour: student’s *t*-test: t = 4.5988, *p* < 0.0001, n = 20; 23-h post binge intake: t = 0.6102, *p* < 0.55, n = 20). Intra-VTA injections of GLP1 also significantly reduced the percentage of daily caloric consumption during the binge hour compared to the saline vehicle (Figure 3B: Mann-Whitney: W = 80.0, *p* < 0.002, n = 20); however, there was no significant difference observed in bout duration between the control verses test groups (Figure 3C: student’s *t*-test: t = 0.1900, *p* < 0.86, n = 20). Additionally, intra-VTA administration of GLP1 significantly reduced meal size (Figure 3D: student’s *t*-test: t = 3.2087, *p* < 0.003, n = 20), and rate of consumption (Figure 3E: student’s *t*-test: t = 2.1520, *p* < 0.04, n = 20) during the binge hour, with no significant impact on meal frequency (Figure 3F: student’s *t*-test: t = 1.5107, *p* < 0.14, n = 20).

To further elucidate the cellular mechanisms underlying the inhibitory effect GLP1 elicits on VTA A_10_ dopamine neurons, we conducted whole-cell patch clamp recordings in A_10_ dopamine neurons from TH-*cre* mice visualized with eYFP (Figure 4A–C). Bath application of GLP1 produced a robust and reversible outward current (Figure 4D), which was nullified when co-administered in the presence of GABA_A_ receptor antagonist SR95531 (Figure 4E). The bath application of GLP1 alongside K_ATP_ channel blocker tolbutamide produced a minor reduction in the outward current seen with administration of GLP1 alone (Figure 4F). Composite data details SR95531 significantly reduced the outward current exhibited with administration of GLP1 (Figure 4G: one-way ANOVA/LSD, *F*: 31.91 DF: 15, *p* < 0.0001: n = 4–8 cells, 4–5 VTA slices, 2–4 animals. Taken together, these data suggest that GLP1 inhibits VTA A_10_ dopamine neurons through GABA_A_ receptor-mediated synaptic input, leading to a decrease in hedonic feeding.

### 2.3. GLP1 Excites VMN PACAP Neurons via Activation of TRPC5 Channels

Now that we have delineated the direct impact of GLP1 on ARC POMC and NPY/AgRP neurons, we next sought to explore the actions of GLP1 following administration into the VMN. Utilizing AgRP-*cre* mice, we performed intra-VMN injections of GLP1 and analyzed homeostatic feeding behavior. VMN administration of GLP1 significantly reduced energy intake 4 & 8 post-delivery (Figure 5A: repeated measures, multifactorial-ANOVA, F_drug_: 38.19, DF: 2, *p* < 0.0001; F_time_: 300.07, DF: 3, *p* < 0.0001; F_interaction_: 2.63, DF: 6, *p* < 0.02; one-way ANOVA/LSD, F: 89.36, DF: 11, *p* < 0.0001, n = 20). While there were no significant changes across treatment groups when analyzing bout duration (Figure 5B: repeated measures, multifactorial-ANOVA: F_drug_: 2.46, DF: 2, *p* < 0.09; F_time_: 1.91, DF:3, *p* < 0.13; F_interaction_: 1.47, DF: 6, *p* < 0.20, n = 20), GLP1 significantly reduced rate of consumption at hours 1 and 4 (Figure 5C: repeated measures, multi-factorial ANOVA: F_drug_: 12.18, DF:2, *p* < 0.0001; F_time_: 11.20, DF: 3, *p* < 0.0001; F_interaction_: 2.96, DF: 6, *p* < 0.009; one-way ANOVA/LSD, F: 6.99, DF: 11, *p* < 0.0001, n = 20). The addition of GLP1 receptor antagonist Exendin 9–39 (30 nmol) reversed the GLP1-induced decreases in energy intake and the rate of consumption at hours 1 and 2 (Figure 5A,C).

We have previously shown that VMN PACAP neurons directly inhibit NPY/AgRP neurons by activating K_ATP_ channels [18]. Utilizing AgRP-*cre*/PAC1R^fl/fl^ mice that we validated previously [18], we again performed intra-VMN injections of GLP1 to determine whether VMN PACAP neurons are possibly the direct targets of GLP1, and responsible for the downstream inhibition of NPY/AgRP neurons via activation of PAC1Rs, thereby promoting loss of homeostatic appetitive behavior. Within the AgRP-*cre*/PAC1R^fl/fl^ cohort, we determined no significant difference between control and GLP1 treated groups in energy intake (Figure 6A: repeated measures multi-factorial ANOVA: F_GLP1_: 2.04, DF: 1, *p* < 0.16; F_time_: 95.43, DF:3, *p* < 0.0001; F_interaction_: 0.65, DF: 3, *p* < 0.59, n = 10–20), nor bout duration (Figure 6B: repeated measures multi-factorial ANOVA: F_GLP1_: 1.67, DF: 1, *p* < 0.20; F_time_: 2.52, DF:3, *p* < 0.07; F_interaction_: 0.42, DF: 3, *p* < 0.74, n = 10–20). GLP1 significantly reduced the rate of consumption during the first hour post GLP1 administration (Figure 6C: repeated measures multi-factorial ANOVA: F_GLP1_: 1.63, DF: 1, *p* < 0.21; F_time_: 4.74, DF: 3, *p* < 0.004; F_interaction_: 2.85, DF: 3, *p* < 0.05; one-way ANOVA/LSD, F: 3.35, DF: 7, *p* < 0.004, n = 10–20), however, this effect was not sustained.

We then performed whole-cell patch clamp recordings in PACAP-*cre* mice to determine the actions of GLP1 on VMN PACAP neurons. We visualized VMN PACAP neurons with eYFP (Figure 7A–C). Bath application of GLP1 produced a robust and reversible inward current in VMN PACAP neurons (Figure 7D). The corresponding I/V plot depicts an increase in slope conductance and a reversal potential of ~−30 mV upon bath application of GLP1 (Figure 7E). Co-administration of GLP1 alongside PI3-kinase inhibitor PI828 (Figure 7F) and PLC inhibitor (Figure 7G) dampened the GLP1-induced robust inward current, and pretreatment with the TRPC5 channel blocker AC1903 30 μM completely reverses this excitatory effect (Figure 7H). Composite data depicts PI828 and U73122 significantly attenuated the change in membrane current; with AC1903 promoting the most significant deviation from the inward current produced by GLP1 (Figure 7I: one-way ANOVA/LSD, *F*: 18.68 DF: 24, *p* < 0.0001: n = 6–9 cells, 3–7 VMN slices, 3–5 animals). The GLP1-induced increase in ion conductance across the membrane was significantly lessened with coadministration alongside PI828, U73122, and AC1903 (Figure 7J: one-way ANOVA/LSD, *F*: 19.41 DF: 24, *p* < 0.0001: n = 6–9 cells, 3–7 VMN slices, 3–5 animals). Retrograde tract tracing coupled with immunohistofluorescence show that GLP1 neurons in the nucleus tractus solitarius project to the VMN (Figure 8A–H), which lends additional physiological credence to these behavioral and electrophysiological findings. Taking these data into consideration, it is evident GLP1 excites VMN PACAP neurons via TRPC5 channels.

### 2.4. VMN PACAP Neurons and PAC1Rs Expressed in A_10_ Dopamine Neurons Play a Relatively Insignificant Role in GLP1-Induced Inhibition of Hedonic Feeding

We have previously established that VMN PACAP neurons project to the VTA and inhibit VTA neurons, including A_10_ dopamine neurons [16]. Given that GLP1 excites VMN PACAP neurons via TRPC5 channels, we sought to investigate whether this excitation disrupts hedonic feeding. We performed intra-VMN administration of GLP1 or its saline vehicle and monitored binge feeding behavior of both TH-*cre* and TH-*cre*/PAC1R^fl/fl^ animals. We found VMN administration of GLP1 had no significant effect on energy intake during the binge hour; however, we noted a significant decrease in energy intake during the 23-h post-binge period (Figure 9A: Binge hour: student’s *t*-test: t = 1.4470, *p* < 0.16, n = 20; 23-h post admin: student’s *t*-test: t = 2.7649, *p* < 0.009, n = 20). In contrast to what we had seen with intra-VTA GLP1 administration, we found no significant change in the percentage of daily caloric consumption during the binge interval with intra-VMN GLP1 (Figure 9B: student’s *t*-test: t = −0.3083, *p* < 0.76, n = 20). GLP1 significantly increased binge duration (Figure 9C: student’s *t*-test: t = −2.2041, *p* < 0.04, n = 20), and elicited no significant change in meal size (Figure 9D: student’s *t*-test: t = 1.2347, *p* < 0.23, n = 20). On the other hand, intra-VMN administration of GLP1 did significantly reduce the rate of consumption during the binge hour (Figure 9E: student’s *t*-test: t = 3.0944, *p* < 0.004, n = 20), and meal frequency remained unchanged (Figure 9F: student’s *t*-test: t = -0.8990, *p* < 0.38, n = 20).

In TH-*cre*/PAC1R^fl/fl^ mice that we characterized and validated previously [19], we found a modest yet significant reduction in energy intake caused by intra-VMN GLP1 during the binge hour and a significant increase in energy intake during the 23-h post-binge period (Figure 10A: Binge hour: student’s *t*-test: t = 2.1396, *p* < 0.04, n _=_ 20–34; 23-h post admin: student’s *t*-test: t = −3.0762, *p* < 0.004, n = 20–34). This corresponded with a reduction in the percentage of daily caloric consumption during the binge period (Figure 10B: student’s *t*-test: t = 3.0841, *p* < 0.004, n = 20–34). The knockdown of PAC1R in A_10_ dopamine neurons negated the GLP-induced increase in bout duration (Figure 10C: student’s *t*-test: t = 0.836743, *p* < 0.41, n = 20–34), and the decrease in the rate of consumption during the binge interval (Figure 10E: student’s *t*-test: t = 1.8852, *p* < 0.07, n = 20–34). Consistent with what we saw in TH-cre mice, intra-VMN GLP1 was without effect on either meal size (Figure 10D: student’s *t*-test: t = 0.3964, *p* < 0.70, n = 20–34) or meal frequency (Figure 10F: student’s *t*-test: t = 1.8324, *p* < 0.08, n = 20–34) in the TH-cre/PAC1R-floxed animals. Taking the data into consideration, it appears that the GLP1-induced excitation of VMN PACAP neurons, and the downstream activation of PAC1Rs in A_10_ dopamine neurons plays a comparatively minor role in regulating hedonic feeding and associated meal pattern.

### 2.5. Caspase Ablation of VMN PACAP Neurons in PACAP-Cre Animals Reduces the GLP1 Induced Reduction in Homeostatic Feeding Behavior

To further determine the role of VMN PACAP neurons in the reduction of homeostatic feeding, we performed intra-VMN AAV injections of caspase-3 as demonstrated and validated previously [18] to focally ablate these cells in PACAP-cre mice. Caspase-3-containing AAV-injected WT mice served as the control. We found a significant reduction in energy intake at hours 4 and 8 in WT mice, and this effect was not observed in PACAP-cre mice (Figure 11A: repeated measures, multifactorial-ANOVA, F_GLP1_: 9.38, DF: 1, *p* < 0.003; F_time_: 130.99, DF: 3, *p* < 0.0001; F_genotype_: 0.05, DF: 1, *p* < 0.83; F_interaction_: 13.06, DF: 3, *p* < 0.0005; one-way ANOVA/LSD, F: 28.73, DF: 15, *p* < 0.0001, n = 20). In WT mice, GLP1 promoted longer bout durations in the WT mice; rendering a significant increase in bout duration at hours 1, 2, and 8, compared to WT vehicle-treated control. By contrast, GLP1 was without effect in PACAP-cre mice (Figure 11B: repeated measures, multifactorial-ANOVA, F_GLP1_: 8.77, DF: 1, *p* < 0.004; F_time_: 0.89, DF: 3, *p* < 0.45; F_genotype_: 20.16, DF: 1, *p* < 0.0001; F_interaction_: 11.59, DF: 3, *p* < 0.0009; one-way ANOVA/LSD, F: 3.16, DF: 15, *p* < 0.0002, n = 20). Additionally, the rate of consumption was significantly decreased in WT mice receiving the GLP1 treatment at hours at all timepoints examined (need stats), but this was not seen in the PACAP-cre mice (Figure 11C: repeated measures, multifactorial-ANOVA, F_GLP1_: 9.70, DF: 1, *p* < 0.003; F_time_: 1.34, DF: 3, *p* < 0.27; F_genotype_: 0.35, DF: 1, *p* < 0.56; F_interaction_: 11.50, DF: 3, *p* < 0.0009; one-way ANOVA/LSD, F: 2.00, DF: 15, *p* < 0.02, n = 20). Taken together, these data demonstrate that VMN PACAP neurons indeed play a significant role in the GLP1 mediated reduction in homeostatic feeding behavior.

## 3. Discussion

In the current study, we assessed the effect of GLP1 in both homeostatic and hedonic feeding. Through intra-ARC, intra-VMN and intra-VTA GLP1 administration, we were able to ascertain the GLP1-influenced decrease in energy intake and binge consumption, and further evaluation in vitro allowed us to tease out neuronal populations and subsequent signaling cascades involved in the observed feeding behaviors. This study elucidates the manner by which GLP1 promotes decreased appetitive behavior under both homeostatic and hedonic conditions, the mechanistic dynamics of which are illustrated by the schematic in Figure 12.

### 3.1. GLP1 Promotes Decreased Feeding Behavior Under Homeostatic Conditions via Excitation of VMN PACAP Neurons While Inhibiting ARC NPY/AgRP Neurons

Our data reveal that both intra-ARC and intra-VMN injection of GLP1 significantly reduced energy intake and rate of consumption in under ad libitum conditions. This is associated, at least in part, with a direct excitation anorexigenic VMN PACAP neurons. The GLP1-induced excitation of POMC neurons has been shown previously to be due to the activation of GLP1 receptors and TRPC5 channels [10,20,21]. However, this marks the first demonstration that GLP1 excites VMN PACAP neurons via the same mechanism that also involves G_q_- and PI3K-mediated signaling. GLP1 receptors are expressed in the VMN [6], and presently we demonstrate that GLP1 neurons in the nucleus tractus solitarius project to the VMN as well. This GLP1-induced excitation in the VMN is similar to that observed for both the adipostat leptin and PACAP itself [22,23,24,25]. Interestingly, the effect of leptin on energy homeostasis and VMN neuronal excitability are nullified by blockade of PAC1 receptors [13,25]. Collectively, this indicates that VMN PACAP neurons may be an important point of convergence for the excitatory actions of leptin, PACAP and GLP1 and their suppression of homeostatic feeding.

Our present results also indicate that GLP1 can dampen homeostatic feeding in part by inhibiting NPY/AgRP neurons through activation of K_ATP_ channels. These data are consistent with those reported by He and coworkers [10], but our current findings delve deeper by demonstrating that the GLP1-induced inhibition these cells requires Gq- and PI3K-mediated signaling for this receptor/effector coupling to take place. The fact that activation of GLP1, PAC1, leptin and insulin receptors expressed in NPY/AgRP neurons all funnel through this same signaling pathway [18,26,27] clearly underscores its integral importance in the regulation of homeostatic feeding. Interestingly, we presently show that the appetite-suppressant effect of intra-VMN GLP1 is abrogated upon knockdown of PAC1 receptors in NPY/AgRP neurons. These findings make for a compelling case for how excitation of VMN PACAP neurons, and subsequent activation of downstream PAC1 receptors that inhibit NPY/AgRP neurons, is a critical feature in the central regulation of homeostatic feeding by GLP1. Indeed, VMN PACAP neurons project to the ARC [17], and NPY/AgRP neurons express PACAP receptors [28]. Activation of PAC1 receptors in NPY/AgRP neurons by PACAP released from upstream VMN neurons inhibits the downstream cell population via KATP channels [18]. The observation that the anorexigenic effect of GLP1 is dependent on the PAC1 receptor-mediated action of PACAP in NPY/AgRP neurons bears a striking resemblance to how the activation of downstream PAC1 receptors is necessary for the anorexigenic effect of leptin within the VMN [13,25].

Thus, it is clear that the reduction in energy intake and rate of consumption observed under homeostatic conditions was resultant of GLP1’s ability to directly excite anorexigenic ARC POMC and VMN PACAP neurons, and directly inhibit orexigenic ARC NPY/AgRP neurons. Taking all data into consideration, the ARC and VMN in the hypothalamus are paramount for the GLP1-mediated decreases in appetitive behavior under homeostatic conditions.

### 3.2. GLP1 Influences Decreased Hedonic Feeding Behavior Through Inducing GABA_A_ Mediated Inhibition of VTA A10 Dopamine Synaptic Output

While we delineated the central effect of GLP1 under homeostatic conditions, it was important to further explore the effects of GLP1 in the hedonic pathway as it is gaining popularity as a therapeutic intervention in treatment of obesity. We found a significant decrease in binge feeding behavior upon intra-VTA administration of GLP1, as well as significant alterations in meal pattern that support this reduction in hedonic consumption. However there was no significant difference in energy intake during the remaining 23-h non-binge feeding parameters. In studying the effects of GLP1 on the hedonic pathway, we found GLP1 inhibits VTA A_10_ dopamine neurons, and this inhibitory effect is significantly attenuated when pretreated with GABA_A_ inhibitor SR95531. In accordance with this finding, others have shown that the GLP1-induced inhibition of ARC NPY/AgRP neurons is due, in part, to enhanced GABA_A_ receptor-mediated input onto these cells [10,20]. Additionally, Wang and coworkers [29] demonstrated NTS GLP1 nerve afferents projecting to the VTA reduce the excitatory synaptic output of VTA dopamine neurons projecting to the nucleus accumbens. Moreover, while our data depict strong suppression of GLP1 on hedonic feeding within the VTA, the GLP1-induced activation of VMN PACAP neurons produced comparatively minor effects in this regard. This indicates that VMN PACAP neurons are not involved to any significant degree in the attenuation of hedonic feeding caused by GLP1. GLP1 binding sites are found in the VTA and nucleus accumbens, and GLP1 neurons in the NTS project to these areas as well [8]. Based on our work as well as that of others, it follows then that GLP1 acts predominately in the VTA and nucleus accumbens to diminish hedonic feeding [4,29,30].

## 4. Materials and Methods

### 4.1. Animal Models

Animal care and procedures were compliant with the NIH Guide for the Care and Use of Laboratory Animals and the Institutional Animal Care and Use Committee at Western University of Health Sciences. This study utilizes tyrosine hydroxylase (TH-*cre*), PACAP-*cre*, POMC-*cre*, and agouti-related peptide (AgRP-*cre*) transgenic mice that were purchased from Jackson laboratories (Stock #008601, #030155, #009593, #012899, respectively). PAC1R^fl/fl^ mice were obtained from Dr. Rachel Ross (Albert Einstein College of Medicine, Bronx, NY, USA), who generated and characterized these mice while at Beth Israel Deaconess Medical Center at Harvard University. TH-*cre* and AgRP-*cre* mice were bred with PAC1R^fl/fl^ mice to produce double transgenic mice. Wildtype mice were bred in house on a C57/BL/6 background. Animals were maintained at 25 °C on a 12 h light–12 h dark cycle (light 06:00–18:00) and provided food and water ad libitum. Once 21 days old, pups were weaned and genotyped using standard PCR protocols. Animals were assigned to either a standard chow diet (Teklad Rodent Diet, Teklad Diets, Madison, WI, USA) with a caloric breakdown as follows: 18% fat, 24% protein, 58% carbohydrates, or a high fat diet (HFD) comprising 45% fat, 20% protein and 35% carbohydrates 5–8 weeks prior to experimentation. Adult male mice from each population (16–35 g; 14–34 weeks old) were used for behavioral and/or electrophysiological experiments.

### 4.2. Surgical Procedures

For certain behavioral experiments mice were surgically outfitted with a 26-gauge guide cannula (Plastics One, Roanoke, VA, USA), while in other behavioral of electrphysiological experiments the subjects underwent stereotaxic injection of adeno-associated viral vector constructs (AAV). Animals were placed in a stereotaxic frame (Stoelting, Wood Dale, IL, USA), and anesthetized with 2% isoflurane. An incision was made to expose the skull and a unilateral or bilateral hole on either side of the mid-sagittal suture was drilled into the VMN (from bregma AP: −0.6 mm; ML: ±0.3 mm; D: −5.6 from dura), ARC (from bregma AP: −0.6 mm; ML: ±0.3 mm, D: −5.9 from dura), or VTA (from bregma AP: −2.1 mm; ±0.5 mm; DV: −4.0 mm from dura). Injections of cre-recombinase dependent AAV1 containing either enhanced yellow fluorescent protein (eYFP) (pAAV-Ef1a-DIO EYFP; 1.0 × 10^13^, 300 nL total volume, Addgene, plasmid #27056, deposited by Karl Deisseroth), or caspase-3 (pAAV-flex-taCasp3-TEVp; 7 × 10^12^ genomic copies/mL); 300 nL total volume (gift from Nirao Shah and Jim Wells; Addgene plasmid #45580) or the retrograde tracer Fluorogold [31] were delivered into either the ARC, VMN or VTA.

### 4.3. Drugs

All drugs were purchased from Tocris Bioscience/R&D Systems (Minneapolis, MN, USA). Electrophysiological experiments utilized the endogenous glucagon-like peptide-1 receptor agonist GLP1 (100 mM stock) diluted with aCSF to a 100 nM working solution, the GABA_A_ receptor antagonist SR95531 (10 mM stock) diluted with aCSF to yield a 10 mM working solution, the K_ATP_ channel blocker tolbutamide prepared as a 100 mM stock solution and further diluted to a 100 mM working concentration with aCSF, tetrodotoxin (TTX) prepared as a 1 mM stock solution in UltraPure H_2_O and further diluted to a working concentration of 500 nM with aCSF, the PI3-kinase inhibitor PI828 was prepared as a 10mM stock solution in DMSO, and further diluted with aCSF to the working concentration of 10μM, the PLC inhibitor U73122 was prepared as a 20 mM stock solution in DMSO, and further diluted with aCSF to the working concentration of 20 μM, the glucagon-like peptide-1 receptor antagonist Exendin-939 (100 mM stock) was diluted with aCSF to a 1 mM working concentration, and the TRPC5 blocker AC1903 was dissolved in DMSO to a stock concentration of 30 mM, and diluted further with aCSF to the working concentration of 30 μM.

For behavioral experiments, GLP-1 and Exendin 9–39 were each dissolved in filtered saline and prepared as a 100 mM stock solution and injected into the ARC, VMN, VTA and VMN respectively. GLP-1 was injected directly into the ARC, VMN, or VTA at a concentration of 30 pmol; 0.2 mL. All aliquots of stock solutions were stored at 4 °C or −20° C until experimentation.

### 4.4. Hypothalamic and Midbrain Slice Preparation

32% isoflurane was utilized to briefly anesthetize the animal prior to rapid decapitation on day of experimentation. The brain was swiftly extracted from the skull and either a coronal hypothalamic or mesencephalic block was procured. The procured block was then mounted on a cutting platform and secured in a vibratome. The vibratome contained an ice cold, oxygenated, sucrose-based cutting solution (NaHCO_3_ 26; dextrose 10, HEPES 10; sucrose 208; KCL 2; NaH_2_PO_4_ 1.25; MgSO_4_ 22; CaCl_2_ 1; in mM). Four coronal slices (300 μm thick) through the ARC or VMN, and two slices (250 μm thick) through the VTA, were obtained. These slices were transferred to an auxiliary chamber that contained room temperature oxygenated aCSF with the following: (NaCl 124; NaHCO_3_ 26; dextrose 10; HEPES 10; KCL 5; NaH_2_PO_4_ 2.6; MgSO_4_ 2; CaCl_2_ 1).

### 4.5. Electrophysiology

Whole-cell patch clamp electrophysiological recordings utilized biocytin-filled electrodes and were performed in hypothalamic slices obtained from male TH-*cre*, AgRP-*cre*, PACAP-*cre* animals. All animals were maintained under ad libitum feeding conditions. During recordings, slices were kept in a chamber which continuously perfused warmed (35 °C), oxygenated aCSF with CaCl_2_ concentration of 2 mM. The aCSF and all drugs diluted with aCSF were perfused at a rate of 1.5 mL/min via a peristaltic pump. Patch electrodes were prepared from borosilicate glass (World Precision Instruments, Sarasota, FL, USA), and filled with internal solution containing the following, in mM: potassium gluconate 128; NaCl 10; MgCl_2_ 1; EGTA 11; HEPES 10; ATP 1; GTP 0.25; 0.5% Biocytin; pH of 7.3 adjusted with KOH; osmolality ranged from 286 to 320 mOsm). Electrode resistances varied from three to eight MΩ.

Recordings were made on an Olympus (Tokyo, Japan) Bx51 W1 fixed stage microscope equipped with infrared differential interference (DIC) video imaging. Multiclamp 700A or 700B preamplifiers (Molecular Devices, LLC, San Jose, CA, USA) served to amplify potentials and pass current through the electrode. Digidata 1550A or 1550B interfaces (Molecular Devices) were coupled to a pClamp 10.5 or 11.2 software, respectively. Digidata was utilized to perform analog to digital conversion of membrane currents and voltages. Access resistance, resting membrane potential (RMP) and input resistance were continuously monitored throughout all recordings. If the access resistance deviated more than 10% from original value, the recording was terminated. Low-pass filtering of currents was conducted at 2 kHz. Liquid junction potential was calculated as 10 mV, and corrected for during data analysis with pClamp software. All voltage clamp recordings were performed under a holding potential of −60 mV.

Recordings were performed in mice injected 2–3 weeks prior to experimentation with an eYFP blank-containing AAV into the ARC (AgRP-*cre*, POMC-*cre*) VMN (PACAP-*cre*), or VTA (TH-*cre*). For all recordings, a baseline current-voltage (I/V) relationship from a holding potential of −60 mV was generated either by administering pulses (10-mV increments; 150 ms duration) which ranged from −140 to −50 mV, or using a ramp protocol (75 mV/s; from −100 to 50 mV). Voltage clamp experiments generated all I/V relationships in the presence of TTX (500 nM). After the baseline I/V, GLP-1 (100 nM) was applied alongside TTX (500 nM), alone or in addition to any one of the following—SR95531 (10 mM), Tolbutamide (100 mM), PI828 (10 μM), AC1903 (30 μM), Exendin 9–39 (1 μM), or U73122 (20 μM), and membrane current was continuously monitored until a new steady-state value was obtained. At this time, a second I/V relationship was generated. During the drug washout, the membrane current was again monitored until it returned to its original baseline value, and a final I/V relationship was generated to ensure reversibility of the induced effect.

### 4.6. Behavioral Studies

Behavioral studies measuring energy intake, bout duration, rate of consumption, and in some instances for binge experiments % daily consumption, meal size, and meal frequency were conducted utilizing a Comprehensive Lab Animal Monitoring System (CLAMS; Columbus Instruments, Columbus, OH, USA) as previously described and validated [16,32,33]. Briefly, energy intake was calculated by multiplying food intake (in g) times the number of calories per unit weight (3.1 kcal/g for standard chow; 4.3 kcal/g for the HFD). Bout duration was determined as the amount of time the animal’s head was over the food trough and actively engaged in energy intake, with a 10-s lower cutoff. The rate of consumption was measured as the energy intake divided by the bout duration in a given hour. The percent daily consumption was ascertained by dividing the energy intake during the binge hour by the total daily energy intake. Meal size refers to the energy intake per bout or meal, whereas meal frequency is the number of meals or bouts per hour. For the homeostatic feeding experiments, standard chow was used throughout. For the binge feeding experiments were given daily, one-hour access to the HFD (from 16:00–17:00 every day over the course of the experiment), and provided ad libitum access to standard chow during the remaining 23 h of the day. For all feeding experiments, animals were injected with either GLP1 (30 pmol; alone or in conjunction with Exendin 9–39 (30 nmol)) or a 0.9% saline vehicle control (0.2 μL) at 16:00 every day over the course of the experiment.

### 4.7. Immunohistochemistry

Slices from wildtype mice were fixed overnight with 4% paraformaldehyde (4% PFM) in Sorenson’s phosphate buffer (pH 7.4). We then immersed them for three days in 20% sucrose dissolved in Sorensen’s buffer, which we replaced daily, followed by snap freezing in 2-methylbutane (EMD Millipore Corporation, Burlington, MA, USA) the next day. Coronal sections (20 μm) through the VMN and VTA were cut on a cryostat and mounted on chilled slides. We then washed these sections with 0.1 M sodium phosphate buffer (PBS; pH 7.4), and then incubated them overnight with either a polyclonal antibody directed against GLP1 (BMA Biomedicals, Rheinstrasse 28–32 CH-4302 Augst Switzerland; 1:800 dilution). The next day we performed two 15-min PBS washes, and then a two-hour application with biotinylated goat anti-rabbit (Jackson ImmunoResearch Laboratories, Inc., West Grove, PA, USA; 1:300 dilution) secondary antibody. After another round of three 15-min PBS washes, we did a last two-hour overlay with streptavidin-Alexa Flour (AF) 488 (Molecular Probes, Inc., Eugene, OR, PA, USA; 1:300), and concluded with a final series of three 30-min PBS washes and cover slipping the slides. Slides were evaluated for Fluorogold and GLP1 colocalization in the NTS via fluorescence immunohistochemistry using a Zeiss Axioskop 2 Plus microscope (Carl Zeiss, Göttingen, Germany) as shown previously [16,31].

### 4.8. Statistical Analysis

Student’s *t*-test was utilized to draw comparisons between two groups. For comparisons made between more than two groups, either one-way or repeated measures, multifactorial analyses of variance (ANOVA) was performed. All analyses were conducted using Statgraphics Centurion Software (version XVI) (The Plains, VA, USA). Differences were considered statistically significant if the alpha probability was <0.05.

## 5. Conclusions

The present study further elucidates the cellular mechanisms by which exogenous GLP1 enacts its appetite suppressant effects via the central nervous system. Under homeostatic conditions, GLP1 suppresses appetite by promoting PACAP release in the ARC via activation of TRPC5 channels in VMN PACAP neurons. Within the bounds of the ARC, GLP1 directly excites POMC neurons and indirectly inhibits NPY/AgRP neurons via as activation of GLP1 receptors, G_q_- and PI3K signaling and K_ATP_ channels. Within the hedonic pathway, GLP1 promotes decreased binge feeding behavior by increasing GABAA receptor-mediated input to A_10_ dopamine neurons. In turn, this suppresses the output from the A_10_ dopamine neurons to the nucleus accumbens of the mesolimbic reward pathway. Taken together, we provide compelling evidence that GLP1 may serve as not only an effective treatment for diabetes management, but also as a promising pharmacotherapy for managing obesity and appetitive behavior.

## Figures and Tables

**Figure 1 ijms-26-03897-f001:**
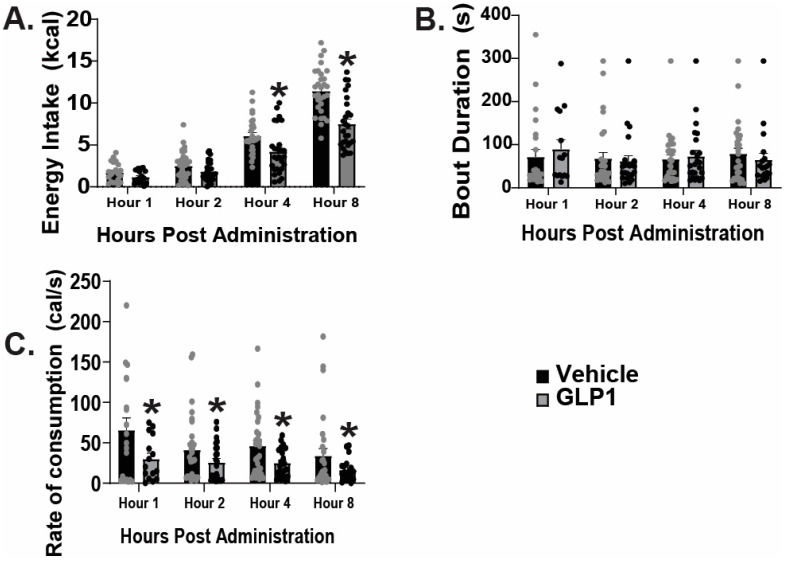
Intra ARC administration of GLP1 significantly reduces homeostatic feeding. Composite data depicting GLP1 (30 pmol; 0.2 μL) significantly reduced energy intake at hours four and eight post intra-ARC administration (**A**), without any effect on bout duration (**B**), and significantly reduced the rate of consumption at all time points examined (**C**). SEM * *p* < 0.05 relative to saline vehicle, n = 30 repeated measures multi-factorial ANOVA.

**Figure 2 ijms-26-03897-f002:**
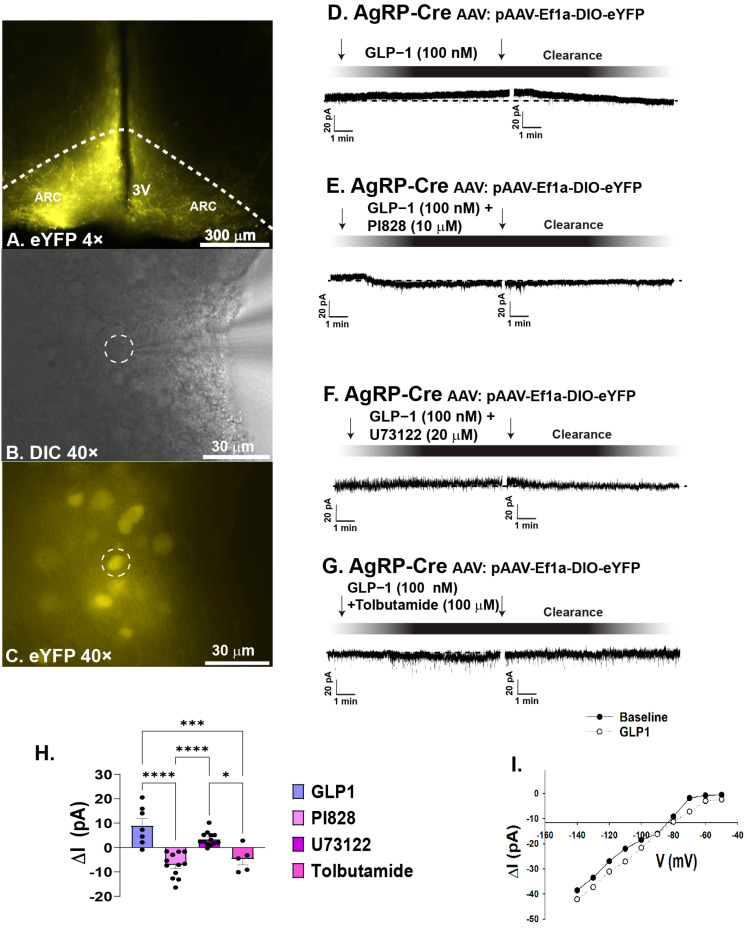
GLP−1 inhibits ARC NPY/AgRP neurons via activation of K_ATP_ channels. (**A**): Low power (4×) image showing YFP-labeled ARC AgRP neurons in a slice from an AgRP-cre mouse, (**B**) high power (40×) DIC image of a recorded AgRP neuron, (**C**) high power (40×) image of the same recorded AgRP neuron labeled with YFP. (**D**): Representative trace depicting the GLP1 induced outward current. This effect is abrogated in the presence of PI3-Kinase inhibitor PI828 (10 μM). (**E**), the PLC inhibitor U73122 (20 μM) (**F**), and the K_ATP_ channel blocker tolbutamide (100 μM) (**G**,**H**): Composite data that depicts the significant reduction in the GLP1-induced outward current by coadministration of GLP1 along with PI828, U73122 or tolbutamide. * *p* < 0.05, *** *p* < 0.001, **** *p* < 0.0001. (**I**): I/V plot illustrating the GLP1−induced increase in slope conductance and the reversal potential of approximately −90, corresponding to the Nernst equilibrium potential for potassium conductance.

**Figure 3 ijms-26-03897-f003:**
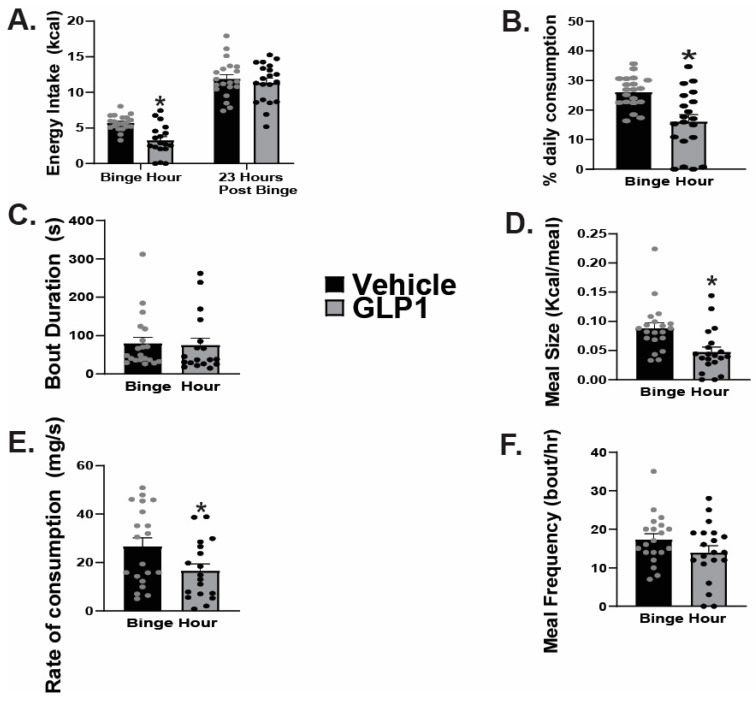
GLP1 delivered into the VTA suppresses hedonic feeding. GLP1 (30 pmol; 0.2 μL) significantly reduces binge consumption with no significant effect on 23-h post-binge intake (**A**), The GLP1-induced decrease in binge consumption is coupled with significant decrements in the % daily caloric consumption during the binge hour (**B**) as well as meal size (**D**) and rate of consumption (**E**). No significant effects on bout duration (**C**) or meal frequency (**F**) were observed. * *p* < 0.05 relative to saline vehicle.

**Figure 4 ijms-26-03897-f004:**
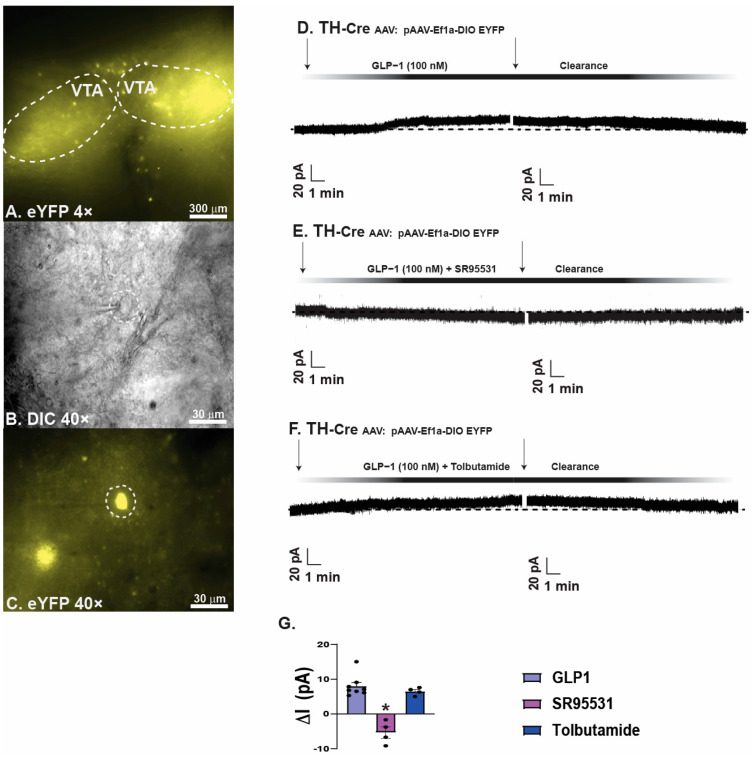
GLP1 inhibits A_10_ dopamine neurons via enhanced GABA_A_ receptor-mediated input onto these cells. (**A**): Low power (4×) image showing YFP-labeled A_10_ dopamine neurons in a slice from a TH-cre mouse, (**B**) high power (40×) DIC image of a recorded A_10_ dopamine neuron depicted in circle, (**C**) high power (40×) image of the same recorded A_10_ dopamine neuron labeled with YFP depicted in circle. (**D**): representative trace showing the robust outward current induced by application of GLP1 (100 nM) in an A_10_ dopamine neurons, and this effect is reversed in the presence of GABA_A_ receptor antagonist SR95531 (10 μM) (**E**), but not in the presence of K_ATP_ channel blocker tolbutamide (100 μM). (**F**,**G**), Composite data depicting that SR95531 promotes a significant change in current compared to GLP1 alone. Bars represent lines and means 1 SEM * *p* < 0.05 relative to GLP1.

**Figure 5 ijms-26-03897-f005:**
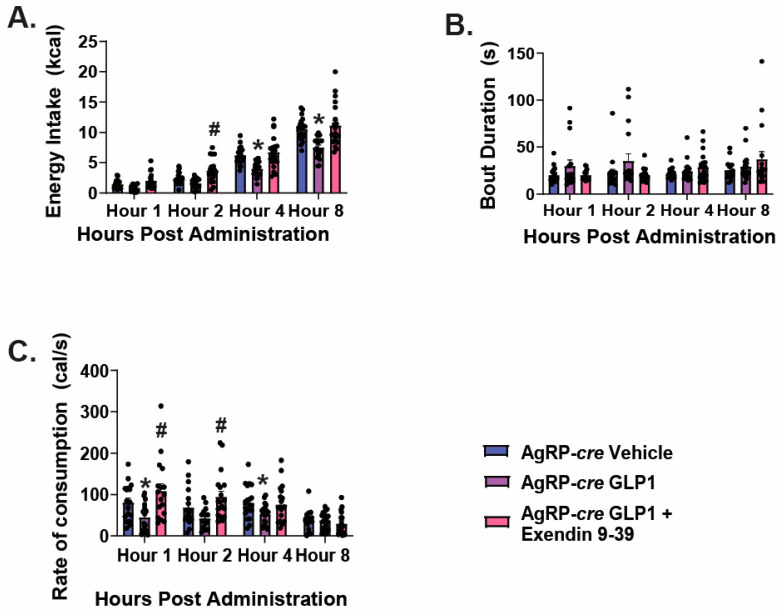
Delivery of GLP1 into the VMN of AgRP-cre mice significantly inhibits energy intake and rate of consumption via activation of GLP1 receptors. (**A**), Composite data depicting intra-VMN administration of GLP1 (30 pmol; 0.2 μL) significantly reduces energy intake at hours 4 and 8. GLP1 had no effect on bout duration (**B**), but it did diminish the rate of consumption out to 4 h post-administration (**C**). These effects are reversed by the GLP1 receptor antagonist exendin 9–39 (30 nmol; 0.2 μL). * *p* < 0.05, GLP1 relative to saline vehicle; # *p* < 0.05, exendin 9–39 relative to saline vehicle.

**Figure 6 ijms-26-03897-f006:**
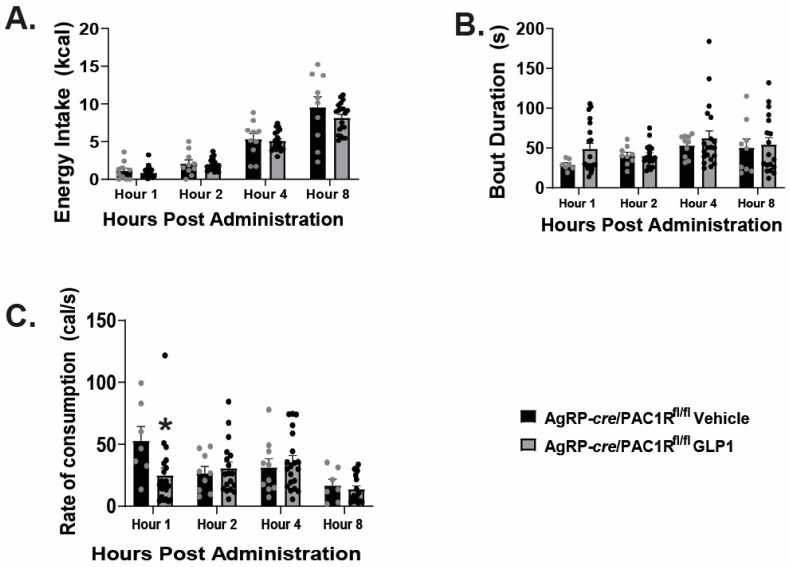
Intra-VMN injections of GLP1 in AgRP-cre PAC1R^fl/fl^ animals largely failed to affect homeostatic feeding and meal pattern. GLP1 (30 pmol; 0.2 μL) had no significant effect on energy intake (**A**) or bout duration (**B**) compared to saline vehicle throughout the longevity of the study. It did, however, elicit a transient decrease in rate of consumption during the first hour post-administration (**C**). * *p* < 0.05 relative to saline vehicle.

**Figure 7 ijms-26-03897-f007:**
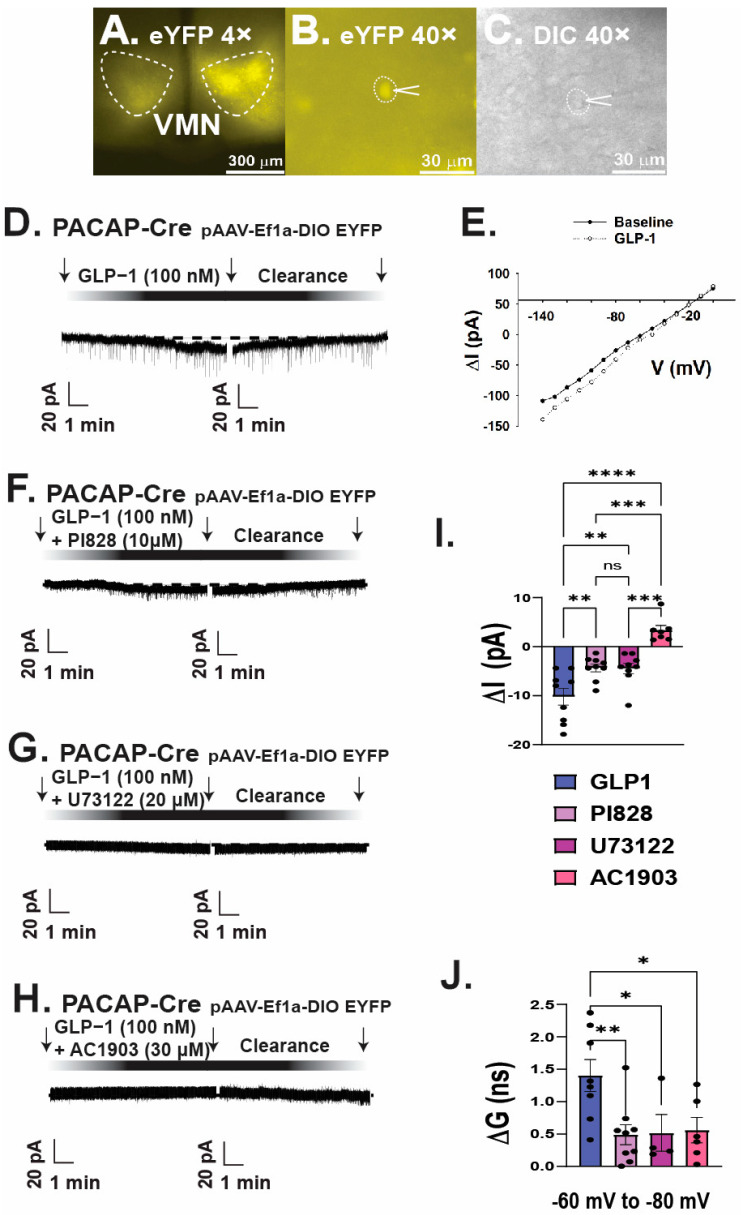
GLP1 excites VMN PACAP neurons via activation of TRPC5 channels. (**A**): Low power (4×) image showing YFP-labeled VMN PACAP-neurons in a slice from a PACAP-cre mouse, (**B**) high power (40×) DIC image of a recorded PACAP neuron, (**C**) high power (40×) image of the same recorded PACAP neuron labeled with YFP. (**D**): Representative trace depicting the excitatory inward current elicited upon bath application of GLP1 (100 nM). (**E**): I/V plot depicting the GLP1-induced increase in slope conductance and reversal potential of approximately −20 mV. These GLP1-induced inward currents are markedly attenuated by PI828 (10 μM) (**F**), U73122 (20 μM). (**G**) and the TRPC5 channel blocker AC1903 (30 μM). (**H**–**J**), Composite data depicting that PI828, U73122, and AC1903 all diminish the GLP-1 induced change in current and slope conductance, respectively. * *p* < 0.05, ** *p* < 0.01, *** *p* < 0.001, **** *p* < 0.0001 relative to GLP1.

**Figure 8 ijms-26-03897-f008:**
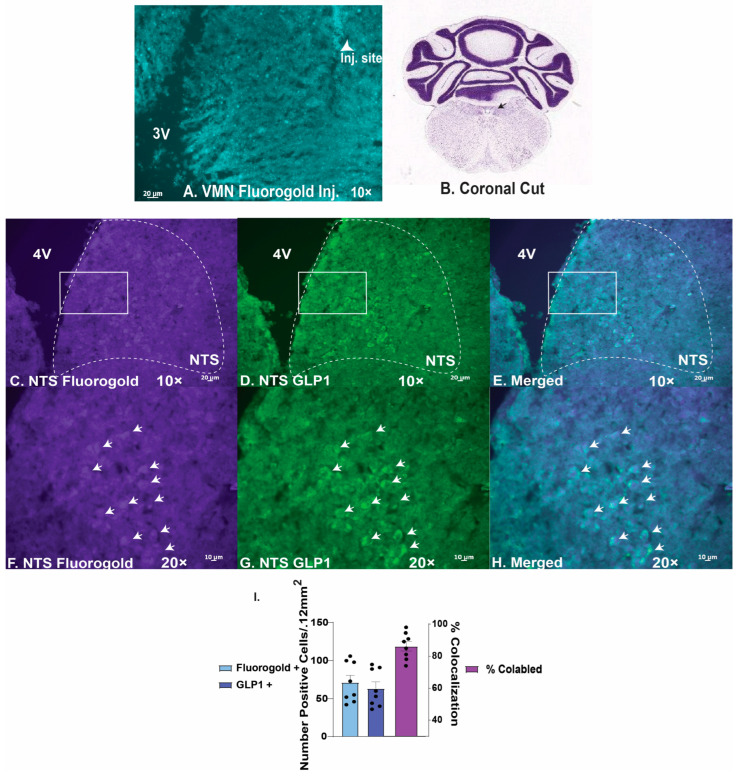
GLP1 neurons in the NTS project to the VMN. (**A**), photomicrograph depicting the Fluorogold injection site in the VMN. (**B**), representative image depicting location of NTS in coronal cut. The middle panel show the Fluorogold-filled neurons (**C**,**F**), the GLP1-positive neurons (**D**,**G**) in the NTS, as well as a merger of the two panels (**E**,**H**) at 10× and 20×, respectively. The bar graph in (**I**) depicts the number of Fluorogold-filled and GLP1-positive NTS neurons, as well as the percentage of neurons colocalizing the two markers.

**Figure 9 ijms-26-03897-f009:**
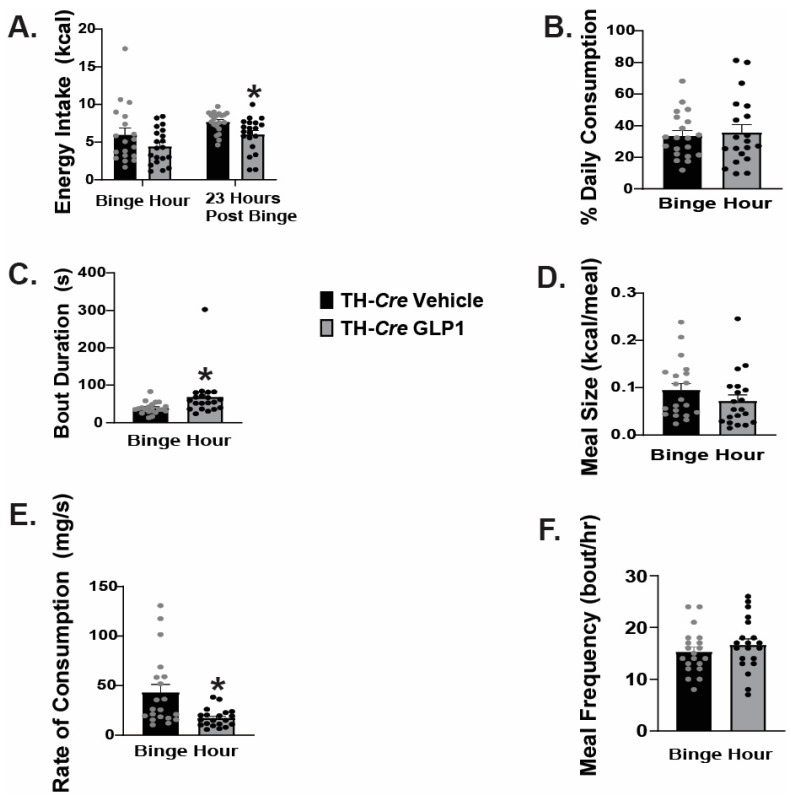
Delivery of GLP1 into the VMN of TH-cre mice significantly reduces homeostatic feeding; with only modest effects on homeostatic feeding. Intra-VMN administration of GLP1 (30 pmol; 0.2 μL) significantly reduces energy intake only during the 23-h post-binge period (**A**). While intra-VMN GLP1 did not affect binge consumption (**A**), the percent daily caloric consumption during the binge hour (**B**), meal size (**D**) or meal frequency (**F**), it did significantly increase bout duration (**C**) and the rate of consumption (**E**). * *p* < 0.05 relative to saline vehicle.

**Figure 10 ijms-26-03897-f010:**
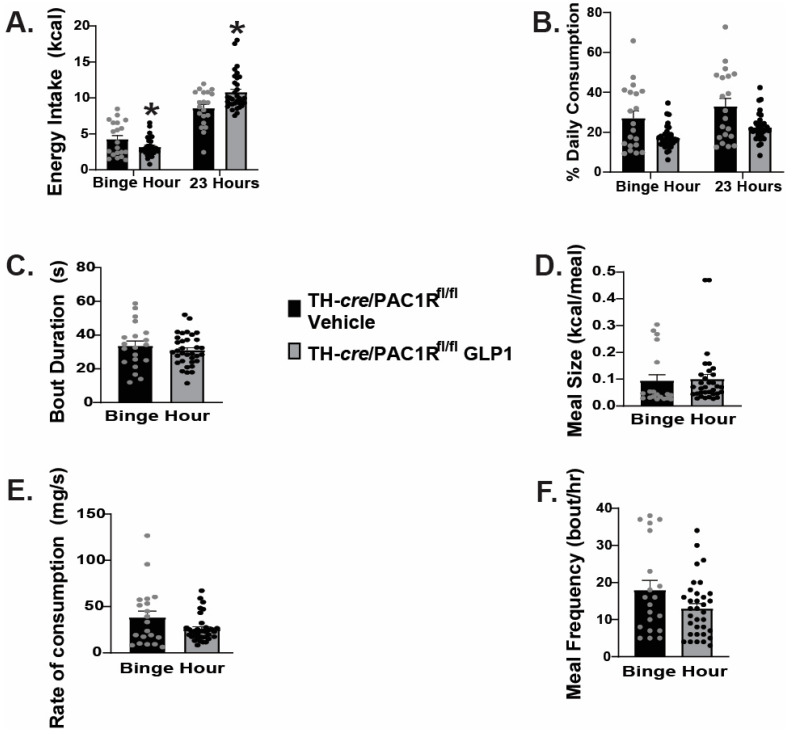
Delivery of GLP1 into the VMN of TH-cre/PAC1R^fl/fl^ mice modestly reduces hedonic feeding while increasing homeostatic feeding. Intra-VMN administration of GLP1 significantly reduces binge consumption, and significantly increases energy intake during the 23-h post-binge period (**A**). The GLP1 (30 pmol; 0.2 μL)-induced decrease in binge consumption is associated with a decrease in the percent daily caloric consumption during the binge hour (**B**) but not bout duration (**C**), meal size (**D**), the rate of consumption (**E**) or meal frequency (**F**). The increased bout duration and decrease rate of consumption caused by GLP1 in TH-cre mice are no longer apparent in TH-cre/PAC1R^fl/fl^ animals. * *p* < 0.05 relative to saline vehicle.

**Figure 11 ijms-26-03897-f011:**
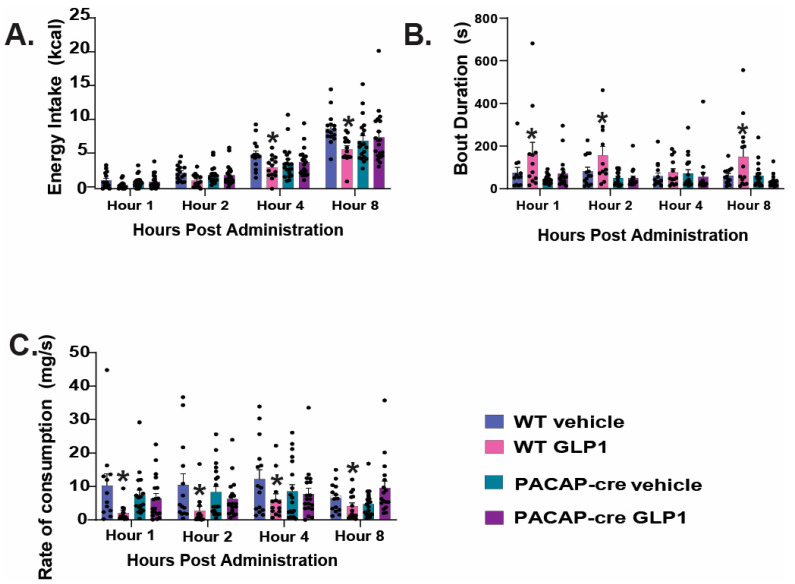
Apoptotic ablation of VMN PACAP neurons with a caspase-3-containing AAV in PACAP-cre mice negates the suppressive effect of GLP-1 homeostatic feeding. (**A**), Composite data depicting intra-VMN administration of GLP1 significantly reduces energy intake at hours 4 and 8 in caspase-3 AAV-injected wildtype but not PACAP-cre mice. GLP1 also increased bout duration in the wildtype mice (**B**), and this was coupled with a decrease in the rate of consumption (**C**). * *p* < 0.05, GLP1 relative to saline vehicle.

**Figure 12 ijms-26-03897-f012:**
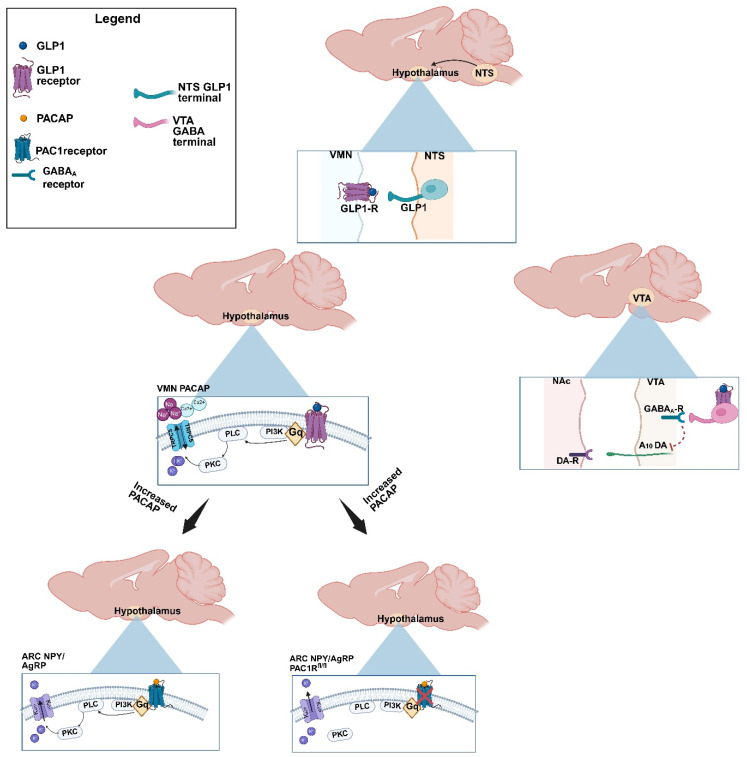
Schematic representation of the mechanisms underlying GLP1 regulation of homeostatic and hedonic feeding. GLP1 neurons in the nucleus tractus solitarius (NTS) project to the VMN. While GLP1 can act within the ARC to inhibit NPY/AgRP neurons, it also excites VMN PACAP neurons via Gq-/PI3K-mediated signaling and activation of TRPC5 channels. VMN PACAP neurons project to the ARC, and the GLP1-induced increase in PACAP release inhibits NPY/AgRP neurons via activation of K_ATP_ channels. These GLP1 receptor-mediated actions account, at least in part, for the suppression of homeostatic feeding caused by the peptide, and are largely abolished by knockdown of PAC1 receptors in NPY/AgRP neurons. Additionally, GLP1 acts within the VTA to inhibit A_10_ dopamine neurons via enhanced GABA_A_ receptor-mediated input onto these cells. This GLP1-induced inhibition of A_10_ dopamine neurons accounts, in part, for the dampening of hedonic feeding caused by the peptide.

## Data Availability

The raw data supporting the conclusions of this article will be made available by the authors, without reservation.

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
