# Peer review of "On the Pleiotropic Actions of Glucagon-like Peptide-1 in Its Regulation of Homeostatic and Hedonic Feeding"

_ijms, 2025, doi:10.3390/ijms26083897_

Round 1
Reviewer 1 Report
Comments and Suggestions for Authors
The manuscript entitled “On the pleiotropic actions of glucagon-like peptide in its regulation of homeostatic and hedonic feeding” by Sayers and Wagner examined the neuroanatomical substrates and signaling mechanisms underlying the suppressive effect of GLP1 on homeostatic and hedonic feeding in mice.
The following are some suggestions to improve the manuscript.
- The authors should provide a brief description of the energy intake studies. This includes the information the diet used for these studies. How long the test mice were starved before running the experiment? How was the change in energy and rate of consumption calculated? What was the effect of intra ARC administration on the weight of mice? They should also explain the terms homeostatic and hedonic feeding frequently used in the manuscript.
- Line 101: It should be Figure 2I in place of 3I.
- Line 103: Please change to: These data depict “that”……
- I understand that the authors injected AAV-EF-1a-DIO-eYFP in AGRP-Cre mice to label AGPR positive neurons for visualization under the microscope for patching, it will be important to mention the same in the results section where the authors are citing Figures 2A-C.
- Line 122: Change to -90 mV
- When was Extendin 9-39 injected in studies presented in Figure 5?
- Description of Figure 7J is missing in the figure legend.
- Individual panel labelling in Figure 8 is not clear. Scale bar information is also not clear. The representative images have a lot of background staining.
- Line 288: Please correct the statement.
- Line 544: The authors have not directly tested the excitability of DA neurons. So, they should modify this statement.
Author Response
Reviewer #1:
1) The authors should provide a brief description of the energy intake studies. This includes the information the diet used for these studies. How long the test mice were starved before running the experiment? How was the change in energy and rate of consumption calculated? What was the effect of intra ARC administration on the weight of mice? They should also explain the terms homeostatic and hedonic feeding frequently used in the manuscript.
Information regarding the diets used in the present study was included in Section 2.1 of the Methods. The animals were never starved or food deprived for any of these experiments. Information regarding how meal duration, rate of consumption and the other parameters were derived is now presented in Section 2.6 of the Methods. Body weights for the experiment mentioned above are included in …, and the terms “homeostatic” and “hedonic feeding are briefly described.
2) Line 101: It should be Figure 2I in place of 3I.
This has been corrected.
3) Line 103: Please change to: These data depict “that” ……
Done.
4) I understand that the authors injected AAV-EF-1a-DIO-eYFP in AGRP-Cre mice to label AGPR positive neurons for visualization under the microscope for patching, it will be important to mention the same in the results section where the authors are citing Figures 2A-C.
Duly noted and taken care of.
5) Line 122: Change to -90 mV
Done.
6) When was Extendin 9-39 injected in studies presented in Figure 5?
At the same time as GLP1. We have clarified this in Section 2.6 of the Methods.
7) Description of Figure 7J is missing in the figure legend.
We have added the missing description. Thank you for pointing this out to us.
8) Individual panel labelling in Figure 8 is not clear. Scale bar information is also not clear. The representative images have a lot of background staining.
Per the reviewer’s suggestion we have modified Figure 8 to provide greater magnification and better clarity to the scale bars and arrows within the individual panels.
9) Line 288: Please correct the statement.
We are not sure of the statement the reviewer refers to in this instance.
10) Line 544: The authors have not directly tested the excitability of DA neurons. So, they should modify this statement.
We have modified this statement as per the reviewer’s suggestion.
Reviewer 2 Report
Comments and Suggestions for Authors
In the manuscript entitled "On the pleiotropic actions of glucagon-like peptide-1 in its regulation of homeostatic and hedonic feeding", the authors investigated the effect of GLP1 brain administration on homeostatic and hedonic appetitive behavior in mice. To this end, they employed gene Cre-Lox conditional knockout mice models, whole-cell patch clamp electrophysiology and brain stereotaxic injections of GLP1 into the Arc, VMH and VTA. The authors demonstrated that GLP1 attenuates feeding in the ARC, partially by inhibiting ARC NPY/AgRP neurons, likely through activation of PI3K-PLC-PKC signaling pathway and KATP channels. Secondly, GLP1 administration into the VTA has been shown to inhibit hedonic feeding by inhibiting VTA A10 dopamine neurons through GABAa receptors. The administration of GLP1 into the VMH has been shown to decrease feeding via GLP1 receptors and/or through the excitation of VMN PACAP neurons via TRPC5 channels. In addition, the authors demonstrated that GLP1 neurons in the NTS project to the VMN. The study further demonstrates that VMN PACAP neurons and PAC1Rs expressed in A10 dopamine neurons play a relatively insignificant role in GLP1-induced inhibition of hedonic feeding. Finally, the study demonstrated that the effect of GLP1 on homeostatic feeding is reduced by the ablation of VMN PACAP neurons. The authors concluded that exogenous GLP1 exerts its effects on homeostatic and hedonic circuits regulating feeding behavior by exciting PACAP neurons in the VMH and inhibiting NPY/AgRP neurons in the Arc and A10 dopamine neurons in the VTA.
The present manuscript provides novel insights into the cellular substrates and molecular mechanisms underlying the central actions of GLP1 in the regulation of homeostatic and hedonic feeding. The study is well designed with a clearly defined objective and methodology. The employment of gene-specific knockout mice models and stereotaxic brain manipulation techniques permitted the authors to study GLP1 brain effects with great precision, thus providing definitive answers to several of their hypotheses. The manuscript is concise and understandable. However, there are some minor concerns that must be addressed prior to the manuscript being considered suitable for publication in IJMS.
Minor concerns:
Do the authors have any explanation for the comparatively modest effects of VMN GLP administration on the feeding of TH-cre mice in comparison to AgRP-cre mice? Could the intrinsic properties of the mouse model (TH-cre) be a reason for the lack of complete GLP effect on hedonic feeding in TH-cre/PAC1R-floxed mice? If this is plausible, it should be mentioned in the Discussion section.
Line 283: Did the authors mean TH-cre/PAC1R-floxed mice?
Line 397: What does NaC stand for? Please provide the full version of the term.
Lines 416-420: The authors should clearly state which type of diet was utilized in each individual experiment.
Line 530: There are two references that need to be resolved.
Author Response
Reviewer #2
1) Do the authors have any explanation for the comparatively modest effects of VMN GLP administration on the feeding of TH-cre mice in comparison to AgRP-cre mice? Could the intrinsic properties of the mouse model (TH-cre) be a reason for the lack of complete GLP effect on hedonic feeding in TH-cre/PAC1R-floxed mice? If this is plausible, it should be mentioned in the Discussion section.
While the AgRP-cre and TH-cre mice come from different strains and backgrounds, we do not think that these differences per se could explain the relative lack of effect of intra-VMN GLP1 on hedonic feeding. Instead, we feel the most parsimonious explanation is that the suppressive of GLP1 on hedonic feeding occurs within the VTA and does not involve VMN PACAP neurons. A statement to this effect has been added to Section 4.2 of the Discussion.
2) Line 283: Did the authors mean TH-cre/PAC1R-floxed mice?
Yes we did. We have corrected this, and apologize for the confusion.
3) Line 397: What does NaC stand for? Please provide the full version of the term.
Per the reviewer’s suggestion we have done so.
4) Lines 416-420: The authors should clearly state which type of diet was utilized in each individual experiment.
The different diets used in the present study were listed in Section 2.1 of the Methods. Further clarity as to which diet(s) were used in the homeostatic vs. hedonic feeding experiments has been provided in Section 2.6.
5) Line 530: There are two references that need to be resolved.
We believe we have fixed the issues related to the references.
We hope the article is now suitable for publication in International Journal of Molecular Sciences.